# Node-Level Differentially Private Graph Neural Networks

## Abstract

Graph Neural Networks (GNNs) are a popular technique for modelling graph-structured data and computing node-level representations via aggregation of information from the neighborhood of each node. However, this aggregation implies increased risk of revealing sensitive information, as a node can participate in the inference for multiple nodes. This implies that standard privacy preserving machine learning techniques, such as differentially private stochastic gradient descent (DP-SGD) – which are designed for situations where each data point participates in the inference for one point only – either do not apply, or lead to inaccurate models. In this work, we formally define the problem of learning GNN parameters with node-level privacy, and provide an algorithmic solution with a strong differential privacy guarantee. We employ a careful sensitivity analysis and provide a non-trivial extension of the privacy-by-amplification technique to the GNN setting. An empirical evaluation on standard benchmark datasets demonstrates that our method is indeed able to learn accurate privacy–preserving GNNs which outperform both private and non-private methods that completely ignore graph information.

## 1 Introduction

Graph Neural Networks (GNNs) (Kipf & Welling, 2016; Veličković et al., 2018; Hamilton et al., 2017; Gilmer et al., 2017) are powerful modeling tools that capture structural information provided by a graph. Consequently, they have become popular in a wide array of domains such as the computational sciences (Ktena et al., 2018; Ahmedt-Aristizabal et al., 2021; McCloskey et al., 2019), computer vision (Wang et al., 2019), and natural language processing (Yao et al., 2019). GNNs have become an attractive solution for modeling users interacting with each other; each user corresponds to a node of the graph and the user-level interactions correspond to edges in the graph. Thus, GNNs are popular for solving a variety of recommendation and ranking tasks, where it is challenging to obtain and store user data (Fan et al., 2019; Budhiraja et al., 2020; Levy et al., 2021). However, such GNN-based solutions are challenging to deploy as they are susceptible to leaking highly sensitive private information of users. Standard ML models – without GNN-style neighborhood data aggregation – are already known to be highly susceptible to leakage of sensitive information about the training data (Carlini et al., 2019). The risk of leakage of private information is even higher in GNNs as each prediction is based on the node itself and aggregated data from its neighborhood. As depicted in Figure 1, there are two types of highly-sensitive information about an individual node that can be leaked in the GNN setting:

- the features associated with the node,

- the labels associated with the node, and,

- the connectivity (relational) information of the node in the graph.

In this work, we study the problem of designing algorithms to learn GNNs while preserving *node*-level privacy.

We use differential privacy as the notion of privacy (Dwork et al., 2006) of a node, which requires that the algorithm should learn roughly similar GNN parameters despite the replacement of an entire node and *all* the data points associated with that node. Our proposed method preserves the privacy of the features of

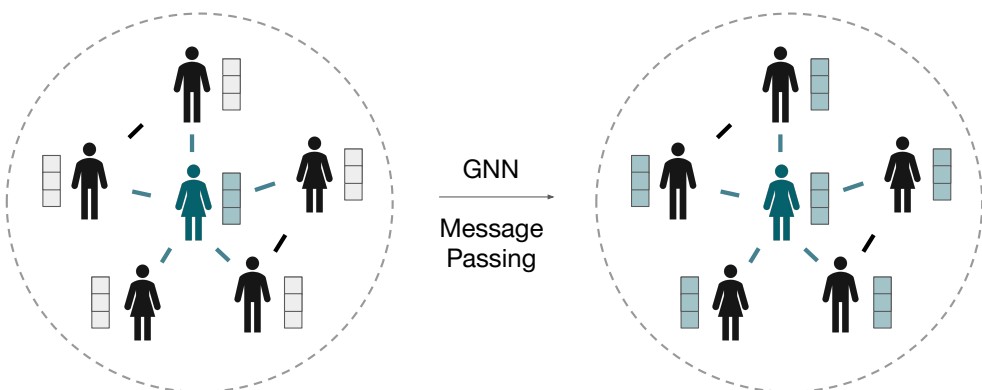

Figure 1: **In a GNN, every node participates in the predictions for neighbouring nodes, introducing new avenues for privacy leakage. The user corresponding to the center node and all of their private information is highlighted in blue.**

each node, their labels as well as their connectivity information. Our method adapts the standard DP-SGD method (Song et al., 2013; Bassily et al., 2014; Abadi et al., 2016) to the node-level privacy setting.

DP-SGD is an extension of standard SGD (stochastic gradient descent) that bounds per-user contributions to the batched gradient by clipping *per-example gradient* terms. However, standard analysis of DP-SGD does not directly extend to GNNs, as each per-example gradient term in GNNs can depend on private data from *multiple nodes*. The key technical contribution of our work is two-fold:

- We propose a graph neighborhood sampling scheme that enables a careful sensitivity analysis for DP-SGD in multi-layer GNNs.

- We extend the standard privacy by amplification technique for DP-SGD in multi-layer GNNs, where one per-example gradient term can depend on multiple users.

Together, this allows us to learn the parameters of a GNN with strong *node*-level privacy guarantees, as evidenced by empirical results on benchmark datasets in Section 6.

## 2 Related Work

Differentially Private SGD (DP-SGD) (Song et al., 2013; Bassily et al., 2014; Abadi et al., 2016) has been used successfully to train neural network models to classify images (Abadi et al., 2016) and text (Anil et al., 2021), by augmenting the standard paradigm of gradient-based training to be differentially private. Edge-level privacy in GNNs ensures that the existence of an edge does not impact the output significantly (Wu et al., 2021b; Epasto et al., 2022). However, such methods do not protect the entirety of each node's private data, and hence provide much weaker privacy guarantees. Private GNNs have also been studied from the perspective of local privacy (Sajadmanesh & Gatica-Perez, 2020; Sajadmanesh et al., 2022), where each node performs its share of the GNN computation locally and sends noisy versions of its data to neighbouring nodes so as to learn shared weights. However, such an algorithm needs to correct for the bias in both the features and labels. Crucially, the analysis of this method only applies to GNNs with linear neighborhood aggregation functions. In contrast, the methods we propose can be employed with a large class of GNN models called 'message-passing' GNNs, described in Section 3. (Wu et al., 2021a) utilizes private GNNs for recommendation systems, but their method assumes a bipartite graph structure, and cannot be naturally extended to homogeneous graphs. Other approaches employ federated learning (Zhou et al., 2020), but only guarantee that the GNN neighbourhood aggregation step is differentially private, which is insufficient to guarantee node-level privacy as we have defined above. Finally, several papers provide privacy-preserving GNNs (Shan et al., 2021) but these do not use the formal notion of DP and provide significantly weaker privacy guarantees. Note that the general concept of node-level differential privacy has been studied before

(Kasiviswanathan et al., 2013; Blocki et al., 2012; Raskhodnikova & Smith, 2016; Karwa et al., 2011; Borgs et al., 2015; 2018). These methods generally estimate global graph-level statistics and do not support learning methods such as GNNs. In contrast, our approach *predicts local* node-level statistics (such as the label of a node) while preserving node-level privacy.

## 3  Problem Formulation and Preliminaries

Consider a graph dataset $G = (V, E, \mathbf{X}, \mathbf{Y})$ with *directed* graph $\mathcal{G} = (V, E)$ represented by a adjacency matrix $\mathbf{A} \in \{0,1\}^{n \times n}$. $n$ is the number of nodes in $\mathcal{G}$, $V$ denotes the node set, $E$ denotes the edge set. Each node $v$ in the graph is equipped with a feature vector $\mathbf{X}_v \in \mathbb{R}^d$; $\mathbf{X} \in \mathbb{R}^{n \times d}$ denotes the feature matrix. $\mathbf{Y} \in \mathbb{R}^{n \times Q}$ is the label matrix and $\mathbf{y}_v$ is the label for the $v$-th node over $Q$ classes. Note that many of the labels in the label vector can be missing, which models the semi-supervised setting. In particular, we assume that node labels $\mathbf{y}_v$ are only provided for a subset of nodes $V_{tr} \subset V$, called the training set. We are interested in predicting labels for the remaining nodes in the graph.

**Traditional Machine Learning Models**: Traditional non-graph based methods such as multi-layer perceptrons (MLPs) perform predictions independently for each node:

$$\widehat{\mathbf{y}}_v = \mathsf{MLP}(\mathbf{X}_v; \mathbf{\Theta}) \tag{1}$$

We see that the prediction by the MLP for node $v$ only depend on the node's feature vector $\mathbf{X}_v$ and the model parameters $\mathbf{\Theta}$.

**Graph Neural Networks**: In contrast, a graph neural network (GNN) uses the relational information between the nodes of the graph, which is captured by the adjacency matrix $\mathbf{A}$. To generate the prediction for a node $v$, the GNN uses features from other nodes $u$ in the local graph neighbourhood of $v$. Each layer in a GNN captures and aggregates features from a larger neighbourhood. Mathematically, a $r$-layer GNN[1] can be generally represented by the following operations:

$$\widehat{\mathbf{y}}_v = \mathsf{GNN}(\mathbf{A}, \mathbf{X}, v; \mathbf{\Theta}) := f_{\text{dec}}\left(f_{\text{agg}}\left(\{f_{\text{enc}}(\mathbf{X}_u) \mid \mathbf{A}_{vu}^r \neq 0\}\right)\right) \tag{2}$$

where $\widehat{\mathbf{y}}_v$ is the prediction from the GNN for a given node $v$, $f_{\text{enc}}$ is the encoder function that encodes node features with parameters $\Theta_{\text{enc}}$, $f_{\text{agg}}$ is the neighborhood aggregation function with parameters $\Theta_{\text{agg}}$, $f_{\text{dec}}$ is the prediction decoder function with parameters $\Theta_{\text{dec}}$, and $\Theta := (\Theta_{\text{enc}}, \Theta_{\text{agg}}, \Theta_{\text{dec}})$. Thus, we can represent an $r$-layer GNN by an aggregation over the local $r$-hop neighborhood of each node. This neighborhood can be represented as a subgraph of $G$ rooted at the node.

Given the graph dataset $G$, the goal is to learn parameters of a GNN while preserving privacy of individual nodes.

While our results apply to most classes of GNN models (Hamilton et al., 2017; Veličković et al., 2018; Xu et al., 2018), for simplicity, we focus on Graph Convolutional Network (GCN) (Kipf & Welling, 2016) with additional results for the Graph Isomorphism Network (GIN) (Xu et al., 2018) and the Graph Attention Network (GAT) (Veličković et al., 2018) in the Appendix. Thus, 'learning' a GNN is equivalent to finding parameters $\Theta := (\Theta_{\text{enc}}, \Theta_{\text{agg}}, \Theta_{\text{dec}})$ that minimize a suitable loss:

$$\Theta^* = \arg\min_{\Theta} \mathcal{L}(G, \Theta) := \sum_{v \in V} \ell(\widehat{\mathbf{y}}_v; \mathbf{y}_v), \tag{3}$$

where $\ell : \mathbb{R}^{Q \times Q} \to \mathbb{R}$ is a standard loss function such as categorical or sigmoidal cross-entropy.

**Definition 1** (Adjacent Graph Datasets)**.** *Two graph datasets $G$ and $G'$ are said to be* **node-level adjacent** *if one can be obtained by adding or removing a node (with its features, labels and associated edges) to the other. That is, $G$ and $G'$ are exactly the same except for the $v$-th node, i.e., $\mathbf{X}_v$, $\mathbf{y}_v$ and $\mathbf{A}_v$ differ in the two datasets.*

---

[1]This is sometimes referred to as a $r$-hop GNN in the literature.

Informally, $\mathcal{A}$ is said to be node-level differentially-private if the addition or removal of a node in $\mathcal{A}$'s input does not affect $\mathcal{A}$'s output significantly.

**Definition 2** (Node-level Differential Privacy (Kasiviswanathan et al., 2013))**.** *Consider any randomized algorithm $\mathcal{A}$ that takes as input a graph dataset. $\mathcal{A}$ is said to be $(\alpha, \gamma)$ **node-level Rényi differentially-private** (Mironov, 2017a) if, for every pair of node-level adjacent datasets $G$ and $G'$: $D_\alpha(\mathcal{A}(G) \parallel \mathcal{A}(G')) \leq \gamma$ where the **Rényi divergence** $D_\alpha$ of order $\alpha$ between two random variables $P$ and $Q$ is defined as $D_\alpha(P \parallel Q) = \frac{1}{\alpha-1} \ln \mathbb{E}_{x \sim Q} \left[ \frac{P(x)}{Q(x)} \right]^\alpha$.*

Note that we use Rényi differentially-private (RDP) (Mironov, 2017a) as the formal notion of differential privacy (DP), as it allows for tighter composition of DP across multiple steps. This notion is closely related to the standard $(\varepsilon, \delta)$-differential privacy (Dwork et al., 2006); Proposition 3 of Mironov (2017a) states that any $(\alpha, \gamma)$-RDP mechanism also satisfies $(\gamma + \frac{\log 1/\delta}{\alpha-1}, \delta)$-differential privacy for any $\delta \in (0,1)$. Thus, we seek to find $\Theta$ by optimizing Equation 3 while ensuring RDP.

**Definition 3.** *The $K$-restricted node-level sensitivity $\Delta_K(f)$ of a function $f$ defined on graph datasets is $\Delta_K(f) = \max_{\substack{G,G' \text{ node-level adjacent} \\ \text{in-deg}(G),\ \text{in-deg}(G') \leq K}} \|f(G) - f(G')\|_2 .$*

**Definition 4.** *We define the **clipping** operator $\text{Clip}_C(.)$ for any vector or matrix $v$ as: $\text{Clip}_C(v) = \min \left( 1, \frac{C}{\|v\|_F} \right) \cdot v.$, where $\|\cdot\|_F$ denotes the Frobenius norm.*

## 4    Sampling Subgraphs with Occurrence Constraints

To bound the sensitivity of the mini-batch gradient in Algorithm 4, we must carefully bound the maximum number of occurrences of any node in the graph across all training subgraphs. To ensure that these constraints are met for any $r$-layer GNN, we propose SAMPLE − SUBGRAPHS (Algorithm 3) to output a set of training subgraphs. In short, we first subsample the edgelists of each node with SAMPLE − EDGELISTS (Algorithm 1), such that each node shows up at maximum $K$ times over all edge lists. Then in RECURSIVE − NEIGHBORHOOD (Algorithm 2), we recursively compute the $r$-depth neighborhood by using the sampled edgelists and the $(r-1)$-depth neighborhood. Note that the 0-depth neighborhood is simply the node itself. This subsampled $r$-depth neighborhood forms the subgraph for each node.

Note that the common practice (Hamilton et al., 2017) of sampling to restrict the out-degree of every node is insufficient to provide such a guarantee, as the in-degree of a node (and hence, the number of occurrences of that node in other subgraphs) can be very large. Once the model parameters have been learnt, no restrictions are needed at inference time. This means GNN predictions for the 'test' nodes can use the entire neighborhood information.

---

**Algorithm 1:** SAMPLE − EDGELISTS: Sampling the Adjacency Matrix with In-Degree Constraints

**Data:** Graph $G = (V, E, \mathbf{X}, \mathbf{Y})$, Training set $V_{tr}$, Maximum in-degree $K$.
**Result:** Set of sampled edgelists $\mathbf{E}_v$ for each node $v \in V$.
**for** $v \in V$ **do**
  Construct the incoming edgelist over training set: $\mathbf{RE}_v \leftarrow \{u \mid (u, v) \in E \text{ and } u \in V_{tr}\}$
  Sample incoming edgelists. Each edge is sampled independently with a probability $p = \frac{K}{2|\mathbf{RE}_v|}$:
   $\mathbf{RE}_v \leftarrow \text{sample}(\mathbf{RE}_v)$
  The nodes with in-degree greater than $K$ are dropped from all edgelists.
**end**
**for** $v \in V$ **do**
  Reverse incoming edgelists to get sampled edgelists $\mathbf{E}_v$: $\mathbf{E}_v \leftarrow \{u \mid v \in \mathbf{RE}_u\}$
**end**
**return** $\{\mathbf{E}_v \mid v \in V\}$.

---

**Lemma 1** (SAMPLE − SUBGRAPHS Satisfies Occurrence Constraints)**.** *Let $G$ be any graph with set of training nodes $V_{tr}$. Then, for any $K, r \geq 0$, the number of occurrences of any node in the set of training*

---

**Algorithm 2:** RECURSIVE − NEIGHBORHOOD: Recursively Computing Node Neighborhoods upto Depth

---

**Data:** Root node $v$, Edgelists $\mathbf{E}$, Maximum depth $r$.
**Result:** Subgraph $S_v$ representing the $r$-depth neighborhood rooted at $v$.
If $r = 0$, **return** $\{v\}$.
Add edges from $v$ to its neighbours' subgraphs:
$\quad S_v \leftarrow \{v\} \cup \{\text{RECURSIVE} - \text{NEIGHBORHOOD}(u, \mathbf{E}, r-1) \mid u \in \mathbf{E}_v\}$
**return** $S_v$.

---

**Algorithm 3:** SAMPLE − SUBGRAPHS: Sampling Local Neighborhoods with Occurrence Constraints

---

**Data:** Graph $G = (V, E, \mathbf{X}, \mathbf{Y})$, Training set $V_{tr}$, Maximum in-degree $K$, GNN layers $r$.
**Result:** Set of subgraphs $S_v$ for each node $v \in V_{tr}$.
Obtain the set of sampled edgelists: $\mathbf{E} \leftarrow \text{SAMPLE} - \text{EDGELIST}(G, V_{tr}, K)$
**for** $v \in V_{tr}$ **do**
$\quad \mid \quad S_v \leftarrow \text{RECURSIVE} - \text{NEIGHBORHOOD}(v, \mathbf{E}, r)$
**end**
**return** $\{S_v \mid v \in V_{tr}\}$.

---

*subgraphs* SAMPLE − SUBGRAPHS$(G, V_{tr}, K, r)$ *is bounded above by* $N(K, r)$, *where:*

$$N(K, r) = \sum_{i=0}^{r} K^i = \frac{K^{r+1} - 1}{K - 1} \in \Theta(K^r)$$

In the interest of space, we have supplied the proof of Lemma 1 in Appendix A.

## 5    Learning Graph Neural Networks (GNNs) via DP-SGD

---

**Algorithm 4:** DP-GNN (SGD): Training Differentially Private Graph Neural Networks with SGD

---

**Data:** Graph $G = (V, E, \mathbf{X}, \mathbf{Y})$, GNN model GNN, Number of GNN layers $r$, Training set $V_{tr}$, Loss function $\mathcal{L}$, Batch size $m$, Maximum in-degree $K$, Learning rate $\eta$, Clipping threshold $C$, Noise standard deviation $\sigma$, Maximum training iterations $T$.
**Result:** GNN parameters $\mathbf{\Theta}_T$.
Note that $V_{tr}$ is the subset of nodes for which labels are available (see Paragraph 1 of Section 3).
Construct the set of training subgraphs with Algorithm 3: $\mathcal{S}_{tr} \leftarrow \text{SAMPLE} - \text{SUBGRAPHS}(G, V_{tr}, K, r)$.
Initialize $\mathbf{\Theta}_0$ randomly.
**for** $t = 0$ **to** $T$ **do**
$\quad \mid \quad$ Sample set $\mathcal{B}_t \subseteq \mathcal{S}_{tr}$ of size $m$ uniformly at random from all subsets of $\mathcal{S}_{tr}$.
$\quad \mid \quad$ Compute the update term $\mathbf{u}_t$ as the sum of the clipped gradient terms in the mini-batch $\mathcal{B}_t$:
$\quad \mid \quad \quad \mathbf{u}_t \leftarrow \sum_{S_v \in \mathcal{B}_t} \text{Clip}_C(\nabla_{\mathbf{\Theta}} \ell (\text{GNN}(\mathbf{A}, \mathbf{X}, v; \mathbf{\Theta}_t); \mathbf{y}_v))$
$\quad \mid \quad$ Add independent Gaussian noise to the update term: $\tilde{\mathbf{u}}_t \leftarrow \mathbf{u}_t + \mathcal{N}(0, \sigma^2 \mathbb{I})$
$\quad \mid \quad$ Update the current estimate of the parameters with the noisy update: $\mathbf{\Theta}_{t+1} \leftarrow \mathbf{\Theta}_t - \frac{\eta}{m} \tilde{\mathbf{u}}_t$
**end**

---

In this section, we describe a variant of DP-SGD (Bassily et al., 2014) designed specifically for GNNs, and show that our method guarantees node-level DP (Definition 2). Assuming we are running a $r$-layer GNN, we first subsample the local $r$-hop neighborhood of each node to ensure that each node has a bounded number of neighbors and influences a small number of nodes. Next, similar to the standard mini-batch SGD technique, we sample a subset $\mathcal{B}_t$ of $m$ subgraphs chosen uniformly at random from the set $\mathcal{S}_{tr}$ of training subgraphs. In contrast to the standard mini-batch SGD, that samples points with replacement for constructing a mini-batch, our method samples mini-batch $\mathcal{B}_t$ uniformly from the set of all training subgraphs. This distinction is important for our privacy amplification result. Once we sample the mini-batch, we apply the standard

DP-SGD procedure of computing the gradient over the mini-batch, clipping the gradient and adding noise to it, and then use the noisy gradients for updating the parameters.

However, DP-SGD requires each update to be differentially private. In standard settings where each gradient term in the mini-batch corresponds to only one point, we only need to add $O(C)$ noise – where $C$ is the clipping norm of the gradient – to ensure privacy. However, in the case of GNNs with node-level privacy, perturbing one node/point $\widehat{\mathbf{v}}$ can have impact on the loss terms corresponding to all its neighbors. Thus, to ensure the privacy of each update, we add noise according to the sensitivity of aggregated gradient: $\nabla_{\mathbf{\Theta}} \mathcal{L}(\mathcal{B}_t; \mathbf{\Theta}_t) := \sum_{S_v \in \mathcal{B}_t} \mathrm{Clip}_C(\nabla_{\mathbf{\Theta}} \ell (\mathsf{GNN}(\mathbf{A}, \mathbf{X}, v; \mathbf{\Theta}_t); \mathbf{y}_v))$ with respect to any individual node $\widehat{\mathbf{v}}$, which we bound via careful subsampling of the input graph. In traditional DP-SGD, a crucial component in getting a better privacy/utility trade-off over just adding noise according to the sensitivity of the minibatch gradient, is privacy amplification by sampling (Kasiviswanathan et al., 2008; Bassily et al., 2014). This says that if an algorithm $\mathcal{A}$ is $\varepsilon$-DP on a data set $D_1$, then on a random subset $D_2 \subseteq D_1$ it satisfies roughly $\frac{|D_2|}{|D_1|} (e^\varepsilon - 1)$-DP. Unlike traditional ERM problems, we cannot directly use this result in the context of GNNs. The reason is again that on two adjacent data sets, multiple loss terms corresponding to $\widehat{\mathbf{v}}$ and its $r$-hop neighbors $\mathcal{N}_{\widehat{\mathbf{v}}}^{(r)}$ get modified. To complicate things further, the minibatch $\mathcal{B}_t$ that gets selected may only contain a small random subset of $\mathcal{N}_{\widehat{\mathbf{v}}}^{(r)}$. To address these issues, we provide a new privacy amplification theorem (Theorem 1). To prove the theorem, we adapt (Feldman et al., 2018, Lemma 25) – that shows a weak form of convexity of Rényi divergence – for our specific instance, and provide a tighter bound by exploiting the special structure in our setting along with the above bound on sensitivity.

**Theorem 1** (Amplified Privacy Guarantee for any $r$-Layer GNN)**.** *Consider the loss function $\mathcal{L}$ of the form:*

$$\mathcal{L}(G, \mathbf{\Theta}) = \sum_{v \in V_{tr}} \ell \left(\mathsf{GNN}(\mathbf{A}, \mathbf{X}, v; \mathbf{\Theta}_t); \mathbf{y}_v\right).$$

*Recall, $N$ is the number of training nodes $V_{tr}$, $K$ is the maximum in-degree of the input graph, $r$ is the number of GNN layers, and $m$ is the batch size. For any choice of the noise standard deviation $\sigma > 0$ and clipping threshold $C$, every iteration $t$ of Algorithm 4 is $(\alpha, \gamma)$ node-level Rényi DP, where:*

$$\gamma = \frac{1}{\alpha - 1} \ln \mathbb{E}_\rho \left[\exp\left(\alpha(\alpha - 1) \cdot \frac{2\rho^2 C^2}{\sigma^2}\right)\right], \ \rho \sim \mathrm{Hypergeometric}\left(N, \frac{K^{r+1} - 1}{K - 1}, m\right).$$

Hypergeometric *denotes the standard hypergeometric distribution (Forbes et al., 2011). By the standard composition theorem for Rényi Differential Privacy (Mironov, 2017a), over $T$ iterations, Algorithm 4 is $(\alpha, \gamma T)$ node-level Rényi DP, where $\gamma$ and $\alpha$ are defined above.*

In the interest of space, we have supplied the proof of Theorem 1 in Appendix B.

**Remark 1**: Roughly, for a 1-layer GNN with $m \gg K$, the above bound implies $\sigma = O(mK/N)$ noise to be added per update step to ensure Rényi DP with $\alpha = O(1)$ and $\gamma = O(1)$. Note that the standard DP-SGD style privacy amplification results do not apply to our setting as each gradient term can be impacted by multiple nodes.

**Remark 2**: We provide node-level privacy, which means that our method preserves the neighborhood information of every node. But, we require a directed graph structure, so that changing a row in the adjacency matrix does not impact any other part of the matrix. This is a natural assumption in a variety of settings. For example, when the graph is constructed by 'viewership' data in social networks, following the rule that edge $(v, v')$ exists iff user $v$ viewed a post from user $v'$.

**Remark 3**: The careful reader may notice that the sampling procedure ensures that the number of gradient terms affected by the removal of a single node is bounded. A natural question to ask is then, can we directly use group privacy guarantees provided by RDP for our analysis? The answer is yes; however, the resulting bounds are much weaker, because the privacy guarantee of group privacy scales exponentially with the size of the group (Mironov, 2017b), which is $N(K, r)$ here. In comparison, Theorem 1 guarantees that the privacy guarantee scales only linearly with $N(K, r)$.

**Remark 4**: For a similar reason as Remark 3, and since privacy amplification by subsampling requires uniform sampling over all nodes (and their corresponding subgraphs) and not edges, one cannot use group privacy guarantees with edge-level differentially private GNNs to obtain an equivalent of Theorem 1.

**Remark 5**: We similarly adapt a DP version of the Adam (Kingma & Ba, 2014) optimizer to the GNN setting, called DP-GNN (Adam), with the same privacy guarantees as DP-GNN (SGD).

**Computational Complexity**: The computational cost and memory requirements of Algorithm 4 is similar to standard DP-SGD which requires computing per-example gradients which are then clipped. Computation of these gradients slows down training by approximately 10 times (Subramani et al., 2020) compared to the base model. Accelerating DP training is an active area of research (Bu et al., 2021).

**Privacy at Inference Time**: Theorem 1 guarantees that the GNN parameters $\Theta$ that are learnt via Algorithm 4 preserve privacy. However, unlike standard ML models where prediction for each point depends only on the model parameters $\Theta$ and the point itself, the privacy of $\Theta$ does not imply that inference using a GNN model will be privacy preserving. In general, the inference about node $v$ can reveal information about its neighbors $\mathcal{N}_v$. Broadly, there are two settings where we can infer labels for a given node while preserving privacy:

1. **Transductive Setting**: Each node has access to the features of its neighbors. In this setting, the aggregation of features from the neighbors does not lead to any privacy loss. Several real-world problems admit such a setting: for example, in social networks where any user has access to a variety of activities/documents/photos of their friends (neighbors). See Section 6.1 for a description of the exact setting we consider for our empirical studies.
2. **Inductive Setting**: Training and test graph datasets are disjoint. In this setting, the goal is to privately learn $\Theta$ using the training graph, that can be 'transferred' to the test graphs. Additionally, the feature information is shared publicly within test graph dataset nodes. A variety of problems can be modeled by this setting: organizations can be represented by a graph over its employees, with the goal to learn a private ranking/recommendation model that can easily be adapted for completely distinct organizations. Section 6.2 discusses empirical studies in this setting.
3. Node features are completely private. In this setting, a node $v$ does not have direct access to the features of its neighbors $\mathcal{N}_v$. Here, the standard GCN model is not directly applicable, but we can still apply GCNs by aggregating the neighborhood features with *noise*. Generally, the resulting prediction for a node would be meaningful only if the degree of the node is reasonably large.

## 6 Experimental Results

In this section, we present empirical evaluation of our method on standard benchmark datasets for large graphs from the Open Graph Benchmark (OGB) suite (Hu et al., 2020) and GraphSAGE (Hamilton et al., 2017), and evaluate our method in both transductive and inductive settings. The goal is to demonstrate that our method (DP-GNN) can indeed learn privacy preserving GNNs accurately. In particular, we benchmark the following methods:

- **DP-GCN**: Our DP-GNN method (Algorithm 4) applied to a 1-layer GCN (in the transductive and inductive settings) and a 2-layer GCN (in the inductive settings) with an MLP as the encoder and the decoder.

- **GCN**: A 1-layer GCN (in transductive and inductive settings) and a 2-layer GCN (in inductive settings) with MLP as the encoder and decoder. In general, this non-private GCN model bounds the maximum accuracy we can hope to achieve from our DP-GCN model.

- **MLP**: A standard multi-layer perceptron (MLP) architecture on the raw node features as proposed in prior works (Hu et al., 2020), which does not utilize any graph information.

- **DP-MLP**: A DP version of a standard MLP trained using DP-Adam.

**Sampling Input Graphs:** Theorem 1 requires an upper bound $K$ on the in-degree of the training nodes in the input graph $G$. For this reason, we subsample the input graph $G$ with Algorithm 1. In Section 6.4, we show that the performance of DP-GNN models is not overly sensitive to the choice of $K$.

**Gradient Clipping:** For DP-GNN and DP-MLP, we perform layer-wise gradient clipping: the gradients corresponding to the encoder, aggregation and decoder functions are clipped independently with different clipping thresholds. For each layer, the clipping threshold $C$ in Algorithm 4 is chosen as the 75th percentile of gradient norms for that layer at initialization on the training data. For simplicity, we perform this estimation of the clipping threshold in a non-private manner, as has been done in previous research (Abadi et al., 2016). We set the noise for each layer $\sigma$ such that the noise multiplier $\lambda = \frac{\sigma}{2C \cdot N(K,r)}$ is identical for each layer, where $\sigma/\lambda$ is essentially the sensitivity. It is not hard to observe that the overall privacy cost only depends on $\lambda$.

**Model Selection:** We select the final model hyperparameters based on the performance on the validation set (separate from the test set). Training was performed until the privacy budget was exhausted. We report the mean and standard deviation of the final model performance on the test set over nine independent runs in Table 1 and Table 2 in the transductive and inductive settings respectively.

Currently, practitioners can not use sensitive graph information in data-critical scenarios, and have to completely discard GNN-based models due to *privacy concerns*. Below, we demonstrate that DP-GNN is able to provide *more accurate solutions* than standard methods that completely discard the graph information, while guaranteeing *node*-level privacy.

## 6.1 Results in the Transductive Setting

Table 1: **Test performance of DP-GCN in the transductive setting, with privacy budget $\varepsilon \leq 10$.**

| Model | ogbn-arxiv Accuracy | ogbn-products Accuracy | ogbn-mag Accuracy | reddit F1-Score |
|---|---|---|---|---|
| GCN (1-layer) | $67.759 \pm 0.394$ | $75.893 \pm 0.479$ | $34.074 \pm 0.42$ | $94.074 \pm 0.07$ |
| **DP-GCN** (Adam) | $61.133 \pm 0.288$ | $67.299 \pm 0.103$ | $28.373 \pm 0.297$ | $91.888 \pm 0.145$ |
| **DP-GCN** (SGD) | $62.382 \pm 0.94$ | $66.698 \pm 0.038$ | $26.785 \pm 0.931$ | $89.474 \pm 0.11$ |
| MLP | $55.236 \pm 0.298$ | $61.157 \pm 0.313$ | $27.121 \pm 0.239$ | $72.347 \pm 0.13$ |
| DP-MLP | $50.803 \pm 0.167$ | $56.006 \pm 0.132$ | $25.26 \pm 0.068$ | $69.041 \pm 0.075$ |

We first study the 'transductive' setting where the features of the test nodes are available during training. At inference time, each node has access to the features of its neighbors. Recall that we focus on 1-layer GNNs in this setting. Table 1 compares the performance of DP-GCN against baselines on the ogbn-arxiv, ogbn-products, ogbn-mag and reddit-transductive datasets. Overall, we observe that our proposed method DP-GCN significantly outperforms the non-private MLP (which does not use any graph information) and private DP-MLP (which does not use any graph information but trained using standard DP-Adam) baselines on all of the datasets and with a privacy budget of $\varepsilon \leq 12$. For example, for ogbn-arxiv and ogbn-products, our method DP-GCN (SGD) is about 6% more accurate than MLP and 10% more accurate than DP-MLP. For reddit, our method is about 20% more accurate than MLP and DP-MLP. Figure 2 provides a comparison of epsilon (privacy guarantee) versus test set performance for the three benchmark datasets. As the privacy budget increases in Figure 2, the performance gap between DP-GCN and the baseline MLP and DP-MLP widens. On all datasets, DP-GCN (Adam) outperforms both MLP and DP-MLP for a privacy budget of $\varepsilon \leq 12$.

Typically, for training non-convex learning models with user-level DP, $\varepsilon \leq 10$ has become a popular choice (Papernot et al., 2020; Kairouz et al., 2021). As the problem is more challenging in the case of GNNs – multiple nodes can affect inference for a given node and we intend to protect privacy at the node-level – a higher $\varepsilon$ seems like a reasonable choice to encourage reasonable solutions. Note that our algorithms satisfy stronger Rényi DP properties (Mironov, 2017a), which provide additional protection over traditional $(\varepsilon, \delta)$-DP guarantees.

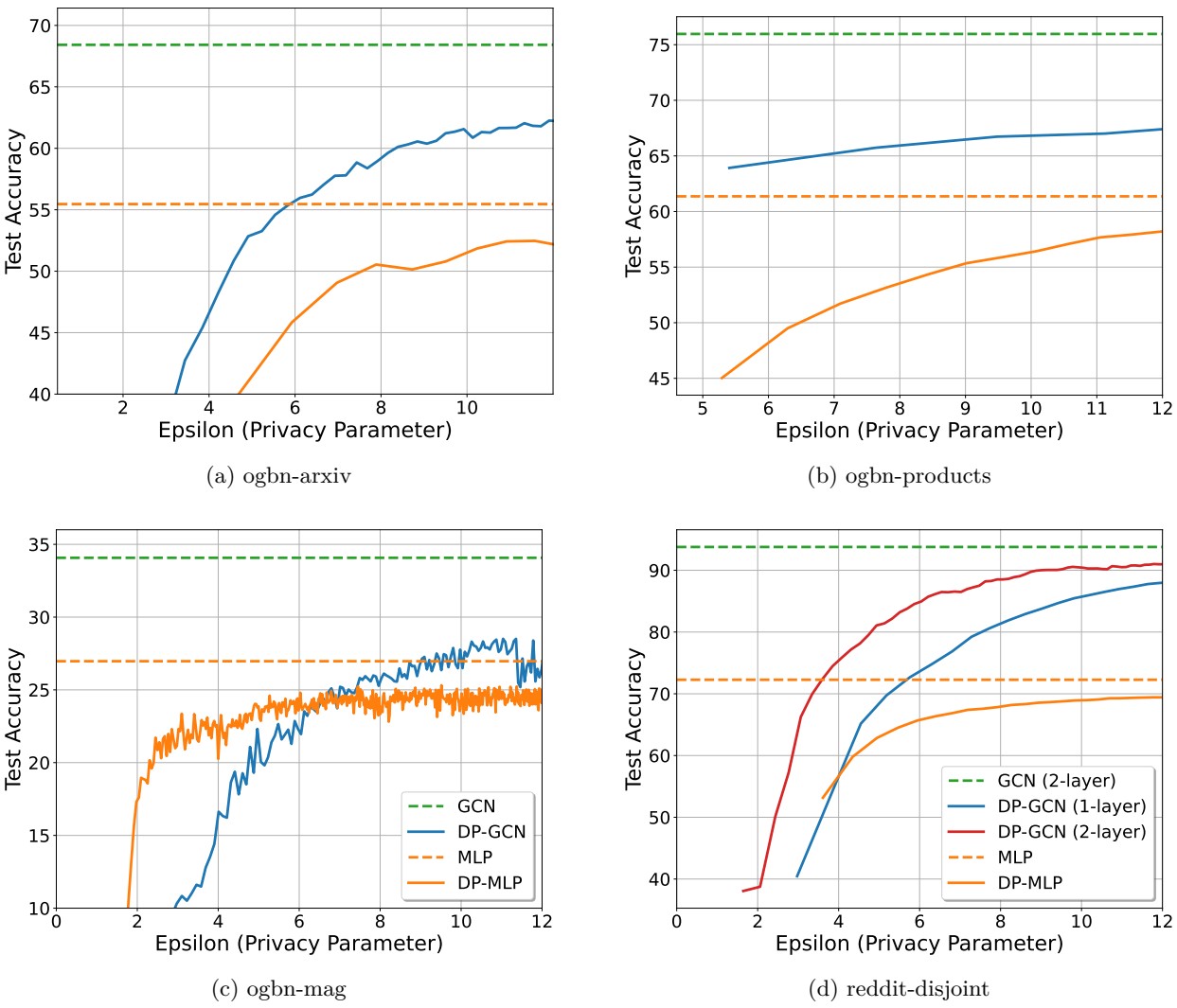

(a) ogbn-arxiv

(b) ogbn-products

(c) ogbn-mag

(d) reddit-disjoint

Figure 2: (a), (b), (c): Performance of the 1-layer DP-GCN models and baselines with respect to privacy budget $\varepsilon$ on ogbn-arxiv, ogbn-products and ogbn-mag datasets. (d): Performance of the 1-layer and 2-layer DP-GCN models on the reddit-disjoint dataset.

## 6.2 Results in the Inductive Setting

Table 2: **Test performance of DP-GCN in the inductive setting, with privacy budget** $\varepsilon \leq 12$**.**

| Model | ogbn-arxiv-disjoint Accuracy | ogbn-arxiv-clustered Accuracy | reddit-disjoint F1-Score |
|---|---|---|---|
| GCN (2-layer) | $60.609 \pm 0.305$ | $56.932 \pm 0.814$ | $93.775 \pm 0.11$ |
| GCN (1-layer) | $60.568 \pm 0.413$ | $54.188 \pm 0.717$ | $92.646 \pm 0.126$ |
| **DP-GCN (2-layer)** | $54.667 \pm 0.182$ | $41.215 \pm 1.081$ | $90.788 \pm 0.124$ |
| **DP-GCN (1-layer)** | $55.155 \pm 0.365$ | $39.713 \pm 0.613$ | $87.711 \pm 0.195$ |
| MLP | $55.349 \pm 0.239$ | $37.764 \pm 0.956$ | $72.272 \pm 0.124$ |
| DP-MLP | $52.27 \pm 0.257$ | $31.639 \pm 1.538$ | $69.541 \pm 0.074$ |

Now, we consider the more challenging 'inductive' setting where the test dataset (the nodes and the graph) are completely disjoint from the training data nodes and the associated graph. This models 'multi-enterprise' situations where the graph over users of one enterprise is completely disjoint from the graph over the users of another enterprise. To conduct these experiments, we divide the nodes into three splits – training, validation and test – and remove all inter-split edges to partition the graph into disjoint subgraphs. We report results on three datasets: ogbn-arxiv-disjoint (where inter-split edges in ogbn-arxiv have been removed), ogbn-arxiv-clustered (where agglomerative clustering[2] is perfomed on the original ogbn-arxiv dataset to partition the nodes) and reddit-disjoint (where inter-split edges in reddit-transductive have been removed). We also investigate 2-layer DP-GCNs in this setting. Once the DP-GNN parameters have been learnt privately over the training graph, we assume that the test graph and test nodes are available non-privately to the inference algorithm. Table 2 presents accuracy of our DP-GNN method with 1-layer and 2-layer GCN models on three datasets. We observe that both 1-layer and 2-layer DP-GCNs are significantly more accurate than MLP and DP-MLP models which completely ignore the graph features.

## 6.3 Results with other GNN Architectures

As mentioned in Section 5, the DP-GNN training mechanisms can be used with most $r$-layer GNN architectures. We experiment with two more GNN architectures, namely GIN (Xu et al., 2018) and GAT (Veličković et al., 2018) in both transductive (Table 3) and inductive (Table 4) settings.

We observe that DP-GNN performs well across different architectures in both privacy settings, outperforming MLP and DP-MLP baselines in all cases. For both the inductive and transductive settings, we observe that GIN performs similarly to GCN, and DP-GIN again has similar performance as DP-GCN. On the ogbn-arxiv-clustered dataset, however, both 1-layer and 2-layer DP-GIN models perform much better than their DP-GCN counterparts.

Table 3: **Test accuracy of DP-GIN and DP-GAT on the transductive ogbn-arxiv dataset with a privacy budget of** $\varepsilon \leq 10$**.**

| Model | Non-Private GNN Accuracy | DP-GNN Accuracy |
|---|---|---|
| GCN | $67.759 \pm 0.394$ | $61.133 \pm 0.288$ |
| GIN | $66.934 \pm 0.498$ | $59.611 \pm 0.383$ |
| GAT | $65.444 \pm 0.465$ | $56.01 \pm 0.303$ |
| MLP | $55.236 \pm 0.298$ | $50.803 \pm 0.167$ |

## 6.4 Ablation Studies

**Batch size** $m$: As has been noted in other DP-SGD works (Abadi et al., 2016; Bagdasaryan et al., 2019; McMahan et al., 2018), we observe that increasing the batch size improves the performance of the learnt

---

[2]See Appendix E for dataset details.

Table 4: **Test performance of DP-GIN in the inductive setting, with privacy budget $\varepsilon \leq 12$.**

| Model | ogbn-arxiv-disjoint Accuracy | ogbn-arxiv-clustered Accuracy | reddit-disjoint F1-Score |
|---|---|---|---|
| GIN (2-layer) | $60.02 \pm 0.657$ | $54.239 \pm 1.699$ | $93.102 \pm 0.199$ |
| GIN (1-layer) | $59.215 \pm 0.313$ | $51.834 \pm 1.284$ | $92.681 \pm 0.18$ |
| **DP-GIN (2-layer)** | $54.966 \pm 0.45$ | $40.658 \pm 1.106$ | $89.342 \pm 0.165$ |
| **DP-GIN (1-layer)** | $56.23 \pm 0.126$ | $41.919 \pm 1.002$ | $89.648 \pm 0.156$ |
| MLP | $55.349 \pm 0.239$ | $37.764 \pm 0.956$ | $72.272 \pm 0.124$ |
| DP-MLP | $52.27 \pm 0.257$ | $31.639 \pm 1.538$ | $69.541 \pm 0.074$ |

Table 5: **Accuracy of GCN and DP-GCN on the ogbn-arxiv dataset with different batch sizes, with DP-GCN privacy budget as $\varepsilon \leq 12$.**

| Batch Size | GCN ($A_{\text{GCN}}$) | DP-GCN ($A_{\text{DP-GCN}}$) | $A_{\text{GCN}} - A_{\text{DP-GCN}}$ |
|---|---|---|---|
| 2500 | 68.809 | 53.514 | 15.295 |
| 5000 | 68.577 | 59.893 | 8.684 |
| 10000 | 68.562 | 62.343 | 6.219 |
| 20000 | 68.393 | 62.995 | 5.398 |
| 40000 | 68.208 | 63.430 | 4.778 |
| Full-Batch | 68.047 | 63.662 | 4.385 |

DP-GNN model. As described in Appendix E, we find that a batch size of 5000 to 10000 to work reasonably well for training DP-GNNs across datasets.

**Maximum In-Degree $K$**: Compared to the batch size $m$, the maximum in-degree $K$ has less of an effect on both non-private and private models trained on ogbn-arxiv, as Table 6 shows. There is still a trade-off: a smaller degree $K$ means lesser differentially private noise added at each update step, but also fewer neighbours for each node to aggregate information from during training. As mentioned previously, this degree constraint only applies to the training nodes. As described in Appendix E, we find that a maximum degree $K$ of around 10 to work reasonably well across datasets.

# 7 Conclusions and Future Work

In this work, we proposed a method to privately learn multi-layer GNN parameters, that outperforms both private and non-private baselines that do not utilize graph information. Our method ensures node-level differential privacy, by a careful combination of sensitivity analysis of the gradients and a privacy amplification result extended to the GNN setting. We believe that our work is a first step in the direction of designing powerful GNNs while preserving privacy. Some promising avenues for future work include extending the DP-GNN method to learn non-local GNNs, guaranteeing inference-time privacy with differentially private neighborhood aggregation, addressing fairness issues, and understanding utility bounds for GNNs with node-level privacy.

Table 6: **Accuracy of GCN and DP-GCN on the ogbn-arxiv dataset with different in-degrees, with DP-GCN privacy budget as $\varepsilon \leq 12$.**

| Degree | GCN ($A_{\text{GCN}}$) | DP-GNN ($A_{\text{DP-GCN}}$) | $A_{\text{GCN}} - A_{\text{DP-GCN}}$ |
|---|---|---|---|
| 3 | 67.931 | 62.413 | 5.518 |
| 5 | 68.003 | 62.830 | 5.173 |
| 7 | 67.622 | 62.884 | 4.738 |
| 10 | 67.615 | 62.534 | 5.081 |
| 20 | 68.185 | 61.506 | 6.679 |
| 30 | 68.241 | 60.528 | 7.713 |

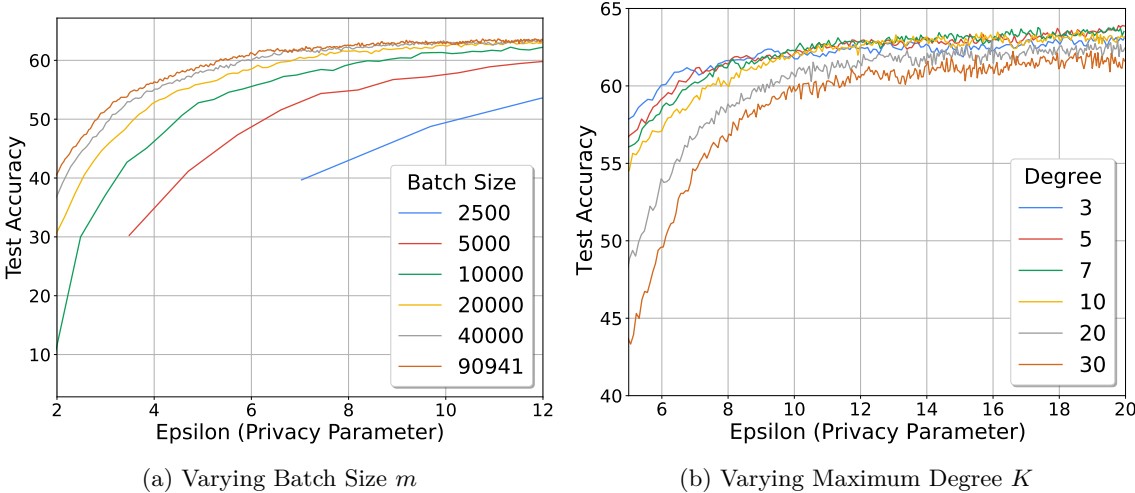

(a) Varying Batch Size $m$          (b) Varying Maximum Degree $K$

Figure 3: **Ablation studies on DP-GCN on the ogbn-arxiv dataset.** (a) shows privacy-utility curves for different batch sizes of DP-GCN, such that the scale of the DP noise added per update step is the same. (b) shows privacy-utility curves for varying maximum in-degree $K$ for the DP-GCN. In both analyses, the other hyperparameters are kept fixed.

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

## Appendix

## A    Proof that SAMPLE − SUBGRAPHS **Satisfies Node Occurrence Constraints**

For clarity, we restate Lemma 1 below.

**Lemma.** *Let $G$ be any graph with set of training nodes $V_{tr}$. Then, for any $K, r \geq 0$, the number of occurrences of any node in the set of training subgraphs* SAMPLE − SUBGRAPHS$(G, V_{tr}, K, r)$ *is bounded above by $N(K, r)$, where:*

$$N(K, r) = \sum_{i=0}^{r} K^i = \frac{K^{r+1} - 1}{K - 1} \in \Theta(K^r)$$

*Proof.* Fix any $K \geq 0$. We proceed by induction on $r$. The proof for $r = 0$, where $S_v$ is simply $\{v\}$, is obvious. In this case, every node $v$ occurs only in its own subgraph $S_v$, and hence, $N(K, r) = \frac{K^1 - 1}{K - 1} = 1$. For any node $v$ and any $r \geq 0$, let $\mathcal{S}^r(v)$ be the set of all subgraphs in SAMPLE − SUBGRAPHS$(G, V_{tr}, K, r)$ in which $v$ occurs. Then, the hypothesis is that $|\mathcal{S}^r(v)| \leq N(K, r)$ for any $v$.

Assume that the given hypothesis holds for some $r$. We will now show that our hypothesis holds for $r + 1$ as well, proving our claim via induction for all $r \geq 0$.

Fix a node $v \in V$. Then, from the definition of RECURSIVE − NEIGHBORHOOD (Algorithm 2), if $S_{u'} \in \mathcal{S}^{r+1}(v)$, then there must exist $u$ such that $u \in \mathbf{E}_{u'}$ and $S_u \in \mathcal{S}^r(v)$. Informally, if $v$ was already in the $r$-depth neighborhood of $u$, and there exists an edge from $u'$ to $u$ after subsampling, then $v$ will be in the $r + 1$-depth neighborhood of $u'$.

By the guarantee of SAMPLE − EDGELISTS (Algorithm 1), the number of nodes such that $u \in \mathbf{E}_{u'}$ is atmost $K$, any node $u$ is present in atmost $K$ edgelists, since its (sampled) in-degree is bounded by $K$. By the inductive hypothesis, there are atmost $N(K, r) - 1$ such nodes $u$ such that $S_u \in \mathcal{S}^r(v)$.

Combining the two upper bounds, and including the subgraph $S_v$ (to which $v$ always belongs), we can derive the upper bound matching the inductive hypothesis for $r + 1$:

$$|\mathcal{S}^{r+1}(v)| \leq N(K, r + 1) = K \cdot (N(K, r) - 1) + 1 = \frac{K^{r+2} - 1}{K - 1}.$$

$\square$

## B    Proof of Privacy Amplification by Subsampling Result for DP-GNN

We provide a detailed proof for Theorem 1 in this section.

**Lemma 2** (Node-Level Sensitivity of any $r$-Layer GNN)**.** *Let $G$ be any graph such that the maximum in-degree of $G$ is bounded by $K \geq 0$. Let $V_{tr}$ be the training set of nodes. Let $\mathcal{B}_t$ be any choice of $m$ unique subgraphs from $\mathcal{S}_{tr} =$ SAMPLE − SUBGRAPHS$(G, V_{tr}, K, r)$. For each node $v \in V_{tr}$, let $\hat{y}_v$ be the prediction from an $r$-layer* GNN *when run on the subgraph $S_v \in \mathcal{S}_{tr}$. Now, $\widehat{\mathbf{y}}_v := $ GNN$(S_v, \mathbf{X}, v; \mathbf{\Theta})$. Consider the loss function $\mathcal{L}$ of the form: $\mathcal{L}(G, \mathbf{\Theta}) = \sum_{v \in V} \ell(\widehat{\mathbf{y}}_v; \mathbf{y}_v)$. Consider the following quantity $\mathbf{u}_t$ from Algorithm 4:*

$$\mathbf{u}_t(G) = \sum_{v \in \mathcal{B}_t} \mathrm{Clip}_C(\nabla_{\mathbf{\Theta}} \ell(\widehat{\mathbf{y}}_v; \mathbf{y}_v))$$

*Then, the following inequality holds:*

$$\Delta_K(\mathbf{u}_t) < 2C \cdot N(K, r) = 2C \cdot \frac{K^{r+1} - 1}{K - 1}.$$

*Proof.* Let $G$ be any graph such that the in-degrees of all nodes in $G$ are bounded by $K \geq 0$. Let $V_{tr}$ be the training set of nodes. Consider a graph $G'$ formed by removing all of the data (features, labels and connections) of a single node $\hat{\mathbf{v}}$ from $G$, so that $G$ and $G'$ are node-level adjacent.

Let us define the two sets of subgraphs from $G$ and $G'$ as follows:

$$\mathcal{S}_{tr} = \mathsf{SAMPLE-SUBGRAPHS}(G, V_{tr}, K, r)$$
$$\mathcal{S}'_{tr} = \mathsf{SAMPLE-SUBGRAPHS}(G', V_{tr}, K, r)$$

Then, the only subgraphs which have changed between $\mathcal{S}_{tr}$ and $\mathcal{S}'_{tr}$ are only those subgraphs in which $\widehat{\mathbf{v}}$ occurred in, that is, $\mathcal{S}^r(v)$ from the proof of Lemma 1. Further, from Lemma 1, there are atmost $N(K, r) = \frac{K^{r+1}-1}{K-1}$ such subgraphs.

We wish to bound the $\ell^2$-norm of the following quantity:

$$\mathbf{u}_t(G) - \mathbf{u}_t(G')$$

For convenience, for any node $v$, we denote the corresponding gradient terms $\nabla_{\boldsymbol{\Theta}} \ell_v$ and $\nabla_{\boldsymbol{\Theta}} \ell'_v$ as:

$$\nabla_{\boldsymbol{\Theta}} \ell_v = \nabla_{\boldsymbol{\Theta}} \ell \left( \mathsf{GNN}(S_v, \mathbf{X}, v; \boldsymbol{\Theta}); \mathbf{y}_v \right) = \nabla_{\boldsymbol{\Theta}} \ell \left( \widehat{\mathbf{y}}_v; \mathbf{y}_v \right)$$
$$\nabla_{\boldsymbol{\Theta}} \ell'_v = \nabla_{\boldsymbol{\Theta}} \ell \left( \mathsf{GNN}(S'_v, \mathbf{X}', v; \boldsymbol{\Theta}); \mathbf{y}_v \right) = \nabla_{\boldsymbol{\Theta}} \ell \left( \widehat{\mathbf{y}}'_v; \mathbf{y}_v \right)$$

Thus, it is clear that the only gradient terms $\nabla_{\boldsymbol{\Theta}} \ell_v$ affected when adding or removing node $\widehat{\mathbf{v}}$, are those corresponding to the subgraphs in $\mathcal{S}^r(v)$:

$$\mathbf{u}_t(G) - \mathbf{u}_t(G') = \sum_{S_v \in (\mathcal{B}_t \cap \mathcal{S}^r(v))} \mathrm{Clip}_C(\nabla_{\boldsymbol{\Theta}} \ell_v) - \mathrm{Clip}_C(\nabla_{\boldsymbol{\Theta}} \ell'_v)$$

In the worst case, all of the subgraphs in $\mathcal{S}^r(v)$ occur in $\mathcal{B}_t$. Each of the gradient terms are clipped to have $\ell^2$-norm $C$. Hence, the triangle inequality gives us:

$$\| \mathrm{Clip}_C(\nabla_{\boldsymbol{\Theta}} \ell_v) - \mathrm{Clip}_C(\nabla_{\boldsymbol{\Theta}} \ell'_v) \|_F \leq \| \mathrm{Clip}_C(\nabla_{\boldsymbol{\Theta}} \ell_v) \|_F + \| \mathrm{Clip}_C(\nabla_{\boldsymbol{\Theta}} \ell_v) \|_F = 2C.$$

Thus:

$$\| \mathbf{u}_t(G) - \mathbf{u}_t(G') \|_F \leq 2C \cdot N(K, r) = 2C \cdot \frac{K^{r+1}-1}{K-1}.$$

The same reasoning applies to the case when $G'$ is formed by a addition of a new node $\widehat{\mathbf{v}}$ to the graph $G$. As $G$ and $G'$ were an arbitrary pair of node-level adjacent graphs, the proof is complete. $\square$

**Lemma 3** (Un-amplified Privacy Guarantee for Each Iteration of Algorithm 4). *Every iteration $t$ of Algorithm 4 is $(\alpha, \gamma)$ node-level Rényi DP where $\gamma = \frac{\alpha \cdot (\Delta_K(\mathbf{u}_t))^2}{2\sigma^2}$.*

*Proof.* Follows directly from Mironov (2017a, Corollary 3). $\square$

**Lemma 4** (Distribution of Loss Terms Per Minibatch). *For any iteration $t$ in Algorithm 4, consider the minibatch $\mathcal{B}_t$ of subgraphs. For any subset $\mathcal{S}$ of $d$ unique subgraphs, define the random variable $\rho$ as $|\mathcal{S} \cap \mathcal{B}_t|$. Then, the distribution of $\rho$ follows the hypergeometric distribution $\mathrm{Hypergeometric}(N, d, m)$:*

$$\rho_i = P[\rho = i] = \frac{\binom{d}{i}\binom{N-d}{m-i}}{\binom{N}{m}},$$

*where $N$ is the number of nodes in the training set $V_{tr}$ and $|\mathcal{B}_t| = m$ is the batch size.*

**Lemma 5** (Adaptation of Lemma 25 from Feldman et al. (2018)). *Let $\mu_0, \ldots, \mu_n$ and $\nu_0, \ldots, \nu_n$ be probability distributions over some domain $Z$ such that: $D_\alpha(\mu_0 \parallel \nu_0) \leq \varepsilon_0, \ \ldots, \ D_\alpha(\mu_n \parallel \nu_n) \leq \varepsilon_n$, for some given $\varepsilon_0, \ldots, \varepsilon_n$.*

*Let $\rho$ be a probability distribution over $[n] = \{0, \ldots, n\}$. Denote by $\mu_\rho$ (respectively, $\nu_\rho$) the probability distribution over $Z$ obtained by sampling $i$ from $\rho$ and then outputting a random sample from $\mu_i$ (respectively, $\nu_i$). Then:*

$$D_\alpha(\mu_\rho \parallel \nu_\rho) \leq \ln \mathbb{E}_{i \sim \rho} \left[ e^{\varepsilon_i(\alpha-1)} \right] = \frac{1}{\alpha-1} \ln \sum_{i=0}^{n} \rho_i e^{\varepsilon_i(\alpha-1)}.$$

**Lemma 6.** *Let $\rho$, $\rho'$ be sampled from the hypergeometric distribution: $\rho \sim \text{Hypergeometric}(N, k, m)$, $\rho' \sim \text{Hypergeometric}(N, k', m)$, such that $k \geq k'$. Then, $\rho$ stochastically dominates $\rho'$: $F_{\rho'}(i) \geq F_\rho(i)$ for all $i \in \mathbb{R}$, where $F_\rho$ (respectively, $F_{\rho'}$) is the cumulative distribution function (CDF) of $\rho$ (respectively, $\rho'$).*

For clarity, we restate Theorem 1 below.

**Theorem** (Amplified Privacy Guarantee for any $r$-Layer GNN)**.** *Consider the loss function $\mathcal{L}$ of the form:*

$$\mathcal{L}(G, \mathbf{\Theta}) = \sum_{v \in V_{tr}} \ell\left(\text{GNN}(\mathbf{A}, \mathbf{X}, v; \mathbf{\Theta}_t); \mathbf{y}_v\right).$$

*Recall, $N$ is the number of training nodes $V_{tr}$, $K$ is the maximum in-degree of the input graph, $r$ is the number of GNN layers, and $m$ is the batch size. For any choice of the noise standard deviation $\sigma > 0$ and clipping threshold $C$, every iteration $t$ of Algorithm 4 is $(\alpha, \gamma)$ node-level Rényi DP, where:*

$$\gamma = \frac{1}{\alpha - 1} \ln \mathbb{E}_\rho \left[ \exp\left( \alpha(\alpha - 1) \cdot \frac{2\rho^2 C^2}{\sigma^2} \right) \right], \quad \rho \sim \text{Hypergeometric}\left( N, \frac{K^{r+1} - 1}{K - 1}, m \right).$$

Hypergeometric *denotes the standard hypergeometric distribution (Forbes et al., 2011). By the standard composition theorem for Rényi Differential Privacy (Mironov, 2017a), over $T$ iterations, Algorithm 4 is $(\alpha, \gamma T)$ node-level Rényi DP, where $\gamma$ and $\alpha$ are defined above.*

*Proof of Theorem 1.* At a high-level, Lemma 2 tells us that a node can participate in $N(K, r) = \frac{K^{r+1} - 1}{K - 1}$ training subgraphs from $\mathcal{S}_{tr}$, in the worst case. However, on average, only a fraction of these subgraphs will be sampled in the mini-batch $\mathcal{B}_t$, as Lemma 4 indicates. We use the knowledge of the exact distribution of the number of subgraphs sampled in $\mathcal{B}_t$ provided by Lemma 4 with Lemma 5 to get a tighter bound on the Rényi divergence between the distributions of $\tilde{\mathbf{u}}_t$ over node-level adjacent graphs. Finally, Lemma 6 allows us to make the above bound independent of the actual node being removed, giving our final result.

Let $G$ be any graph with training set $V_{tr}$. Let $G'$ be formed by removing a single node $\hat{\mathbf{v}}$ from $G$, so $G$ and $G'$ are node-level adjacent.

For convenience, for any node $v$, we denote the corresponding gradient terms $\nabla_{\mathbf{\Theta}} \ell_v$ and $\nabla_{\mathbf{\Theta}} \ell'_v$ as:

$$\nabla_{\mathbf{\Theta}} \ell_v = \nabla_{\mathbf{\Theta}} \ell\left(\text{GNN}(S_v, \mathbf{X}, v; \mathbf{\Theta}); \mathbf{y}_v\right) = \nabla_{\mathbf{\Theta}} \ell\left(\hat{\mathbf{y}}_v; \mathbf{y}_v\right)$$
$$\nabla_{\mathbf{\Theta}} \ell'_v = \nabla_{\mathbf{\Theta}} \ell\left(\text{GNN}(S'_v, \mathbf{X}', v; \mathbf{\Theta}); \mathbf{y}_v\right) = \nabla_{\mathbf{\Theta}} \ell\left(\hat{\mathbf{y}}'_v; \mathbf{y}_v\right)$$

Let $\mathcal{S}^r(v)$ be the set of all subgraphs in $\text{SAMPLE} - \text{SUBGRAPHS}(G, V_{tr}, K, r)$ in which $v$ occurs.

$$\mathbf{u}_t(G) - \mathbf{u}_t(G') = \sum_{S_v \in (\mathcal{B}_t \cap \mathcal{S}^r(\hat{\mathbf{v}}))} \text{Clip}_C(\nabla_{\mathbf{\Theta}} \ell_v) - \text{Clip}_C(\nabla_{\mathbf{\Theta}} \ell'_v).$$

Using the notation from Algorithm 4, we have:

$$\tilde{\mathbf{u}}_t(G) = \mathbf{u}_t(G) + \mathcal{N}(0, \sigma^2 \mathbb{I})$$
$$\tilde{\mathbf{u}}_t(G') = \mathbf{u}_t(G') + \mathcal{N}(0, \sigma^2 \mathbb{I})$$

We need to show that $D_\alpha(\tilde{\mathbf{u}}_t(G) \parallel \tilde{\mathbf{u}}_t(G')) \leq \gamma$.

From the above equation, we see that the sensitivity of $\mathbf{u}_t$ depends on the number of subgraphs in $\mathcal{S}^r(v)$ that are present in $\mathcal{B}_t$. Let $\rho'$ be the distribution over $\{0, 1, \ldots |\mathcal{S}^r(v)|\}$ of the number of subgraphs in $\mathcal{S}^r(v)$ present in $\mathcal{B}_t$, that is, $\rho' = |\mathcal{S}^r(v) \cap \mathcal{B}_t|$. Lemma 4 then gives us that the distribution of $\rho'$ is: $\rho' \sim \text{Hypergeometric}(N, \mathcal{S}^r(v), m)$. In particular, when $\rho' = i$, exactly $i$ subgraphs are sampled in $\mathcal{B}_t$. Then, following Lemma 2, $\Delta_K(\mathbf{u}_t \mid \rho' = i) < 2iC$. Thus, conditioning on $\rho' = i$, we see that every iteration is $(\alpha, \gamma_i)$ node-level Rényi DP, by Lemma 3 where $\gamma_i = \alpha \cdot 2i^2 C^2 / \sigma^2$.

Define the distributions $\mu_i$ and $\nu_i$ for each $i \in \{0, \ldots, |\mathcal{S}^r(v)|\}$, as follows: $\mu_i = [\tilde{\mathbf{u}}_t(G) \mid \rho' = i], \nu_i = [\tilde{\mathbf{u}}_t(G') \mid \rho' = i]$. Then $D_\alpha(\mu_i \parallel \nu_i) \leq \gamma_i$. For the mixture distributions $\mu_{\rho'} = \tilde{\mathbf{u}}_t(G)$ and $\nu_{\rho'} = \tilde{\mathbf{u}}_t(G')$, Lemma 5 now tells us that:

$$D_\alpha(\tilde{\mathbf{u}}_t(G) \parallel \tilde{\mathbf{u}}_t(G')) = D_\alpha(\mu_{\rho'} \parallel \nu_{\rho'})$$
$$\leq \frac{1}{\alpha - 1} \ln \mathbb{E}_{i \sim \rho'} \left[ \exp\left(\gamma_i(\alpha - 1)\right) \right]$$
$$= \frac{1}{\alpha - 1} \ln \mathbb{E}\left[f(\rho')\right].$$

where:

$$f(\rho') = \exp\left(\alpha(\alpha - 1) \cdot \frac{2\rho'^2 C^2}{\sigma^2}\right).$$

Define another distribution $\rho$ as:

$$\rho \sim \mathrm{Hypergeometric}(N, N(K, r), m)$$

where $N(K, r) = \frac{K^{r+1} - 1}{K - 1}$ is the upper bound on $|\mathcal{S}^r(v)|$ from Lemma 1. By Lemma 6, $\rho$ stochastically dominates $\rho'$. As non-decreasing functions preserve stochastic dominance (Hadar & Russell, 1969), $f(\rho)$ stochastically dominates $f(\rho')$, and hence $\mathbb{E}\left[f(\rho')\right] \leq \mathbb{E}\left[f(\rho)\right]$. It follows that:

$$D_\alpha(\tilde{\mathbf{u}}_t(G) \parallel \tilde{\mathbf{u}}_t(G')) \leq \frac{1}{\alpha - 1} \ln \mathbb{E}\left[f(\rho)\right]$$
$$= \frac{1}{\alpha - 1} \ln \mathbb{E}_\rho \left[ \exp\left(\alpha(\alpha - 1) \cdot \frac{2\rho^2 C^2}{\sigma^2}\right) \right] = \gamma.$$

The theorem now follows from the fact that the above holds for an arbitrary pair of node-level adjacent graphs $G$ and $G'$. $\qquad \square$

## C  Class-wise Analysis of Learnt Models

To better understand the performance of the private model as compared to the non-private baseline for our considering setting of multi-class classification at a node-level, we compare the accuracy of these two models for each dataset at a class-wise granularity. These results are illustrated in Figure 4. We empirically observe that the performance of the private model degrades as the frequency of training data points for a particular class decreases. This indicates that the model is able to classify data points of 'frequent' classes with reasonable accuracy, but struggles with classification accuracy on the data points of 'rarer' classes. This observation is in line with previous claims from (Bagdasaryan et al., 2019; Fioretto et al., 2021) that differentially-private models generally perform disparately worse on under-represented classes. While methods (Jagielski et al., 2018; Xu et al., 2021) have been developed for improving the fairness of DP-SGD, their extension to the GNN setting represents an important direction for future work.

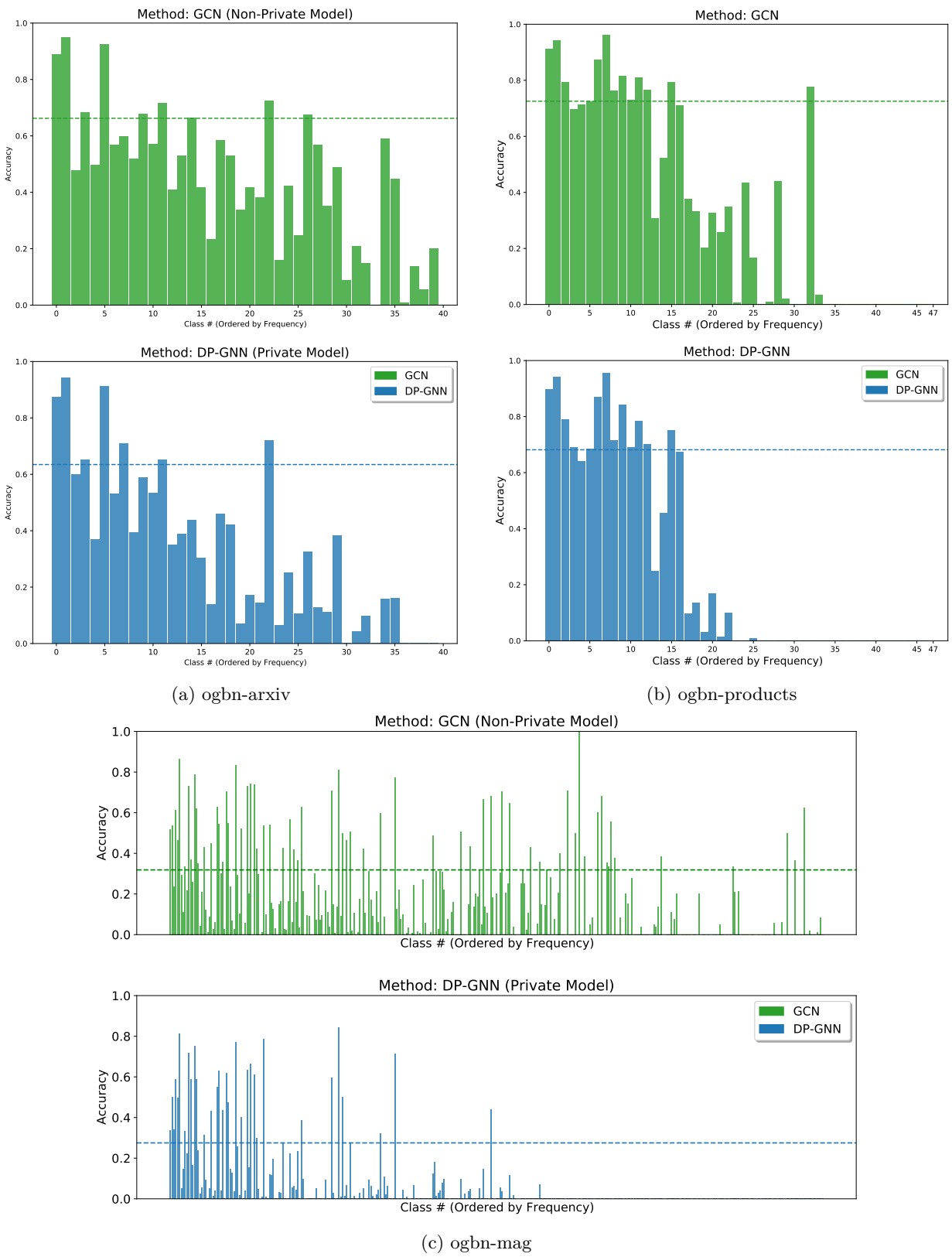

Figure 4: **Comparison of class-wise test accuracies of the non-private GCN model and private DP-GCN model on all datasets, ordered by the decreasing frequency of occurrence of classes in the training data from left to right.** The dotted lines indicate the overall ('micro') accuracy for each model.

## D   Learning Graph Convolutional Networks (GCN) via DP-Adam

In Algorithm 5, we provide the description of DP-Adam, which adapts Algorithm 4 to use the popular Adam (Kingma & Ba, 2014) optimizer, instead of SGD. The privacy guarantee and accounting for Algorithm 5 is identical to that of Algorithm 4, since the DP clipping and noise addition steps are identical.

---

**Algorithm 5:** DP-GNN (Adam): Differentially Private Graph Neural Network with Adam

---

**Data:** Graph $G = (V, E, \mathbf{X}, \mathbf{Y})$, GNN definition GNN, Training set $V_{tr}$, Loss function $\mathcal{L}$, Batch size $m$,
   Maximum degree $K$, Learning rate $\eta$, Clipping threshold $C$, Noise standard deviation $\sigma$,
   Maximum training iterations $T$, Adam hyperparameters $(\beta_1, \beta_2)$.

**Result:** GNN parameters $\mathbf{\Theta}_T$.

Note that $V_{tr}$ is the subset of nodes for which labels are available (see Paragraph 1 of Section 3).

Construct the set of training subgraphs with Algorithm 3: $\mathcal{S}_{tr} \leftarrow \mathsf{SAMPLE} - \mathsf{SUBGRAPHS}(G, V_{tr}, K, r)$.

Initialize $\mathbf{\Theta}_0$ randomly.

**for** $t = 0$ **to** $T$ **do**

 Sample set $\mathcal{B}_t \subseteq \mathcal{S}_{tr}$ of size $m$ uniformly at random from all subsets of $\mathcal{S}_{tr}$.

 Compute the update term $\mathbf{u}_t$ as the sum of the clipped gradient terms in the mini-batch $\mathcal{B}_t$:

$$\mathbf{u}_t \leftarrow \sum_{S_v \in \mathcal{B}_t} \mathrm{Clip}_C(\nabla_{\mathbf{\Theta}} \ell \left( \mathsf{GNN}(\mathbf{A}, \mathbf{X}, v; \mathbf{\Theta}_t); \mathbf{y}_v \right))$$

 Add independent Gaussian noise to the gradient term: $\tilde{\mathbf{u}}_t \leftarrow \mathbf{u}_t + \mathcal{N}(0, \sigma^2 \mathbb{I})$

 Update first and second moment estimators with the noisy gradient, correcting for bias:

$$f_t \leftarrow \beta_1 \cdot f_{t-1} + (1 - \beta_1) \cdot \tilde{\mathbf{u}}_t$$
$$s_t \leftarrow \beta_2 \cdot s_{t-1} + (1 - \beta_2) \cdot (\tilde{\mathbf{u}}_t \odot \tilde{\mathbf{u}}_t)$$
$$\widehat{f}_t \leftarrow \frac{f_t}{1 - \beta_1^t}$$
$$\widehat{s}_t \leftarrow \frac{s_t}{1 - \beta_2^t}$$

 Update the current estimate of the parameters with the noisy estimators:

$$\mathbf{\Theta}_{t+1} \leftarrow \mathbf{\Theta}_t - \frac{\eta}{m} \frac{\widehat{f}_t}{\sqrt{\widehat{s}_t^2} + \varepsilon}$$

**end**

---

## E   Reproducibility

Our open-sourced pipeline for sampling graph datasets and training DP-GNN and the other baseline models is available at (URL to be provided on paper acceptance, currently anonymized and uploaded as supplementary material). All experiments were run on TPU v2 Donuts (2x2 and 4x4 topology).

**Data:** Table 7 provides details about the node classification benchmark datasets used in this work: ogbn-arxiv, ogbn-products, ogbn-mag[3], and reddit[4].

**Models**: We use the following 'inverse-degree' normalization of the adjacency matrix for all GCN models: $\widehat{\mathbf{A}} = (d + \mathbb{I})^{-1}(\mathbf{A} + \mathbb{I})$. We use a variant of the original GAT architecture, utilizing dot-product attention instead of additive attention, with 10 attention heads. Adam (Kingma & Ba, 2014) with $\beta_1 = 0.9$ and $\beta_2 = 0.999$, and SGD optimizers were used for training all methods for each of the datasets. A latent

---

[3]Obtained from `https://ogb.stanford.edu/docs/nodeprop/`.
[4]Obtained from `http://snap.stanford.edu/graphsage/`. Available preprocessed at this Google Drive link.

Table 7: **Statistics of datasets used in our experiments.**

| Dataset | Nodes | Avg. Degree | Features | Classes | Train/Val/Test Split |
|---|---|---|---|---|---|
| ogbn-arxiv | 169,343 | 13.7 | 128 | 40 | 0.54/0.18/0.28 |
| ogbn-arxiv-disjoint | 169,343 | 5.5 | 128 | 40 | 0.54/0.18/0.28 |
| ogbn-arxiv-clustered[5] | 169,343 | 13.0 | 128 | 40 | 0.67/0.18/0.15 |
| ogbn-products | 2,449,029 | 50.5 | 100 | 47 | 0.08/0.02/0.90 |
| ogbn-mag | 736,389 | 21.7 | 128 | 349 | 0.85/0.09/0.05 |
| reddit | 232,965 | 99.6 | 602 | 41 | 0.66/0.10/0.24 |
| reddit-disjoint | 232,965 | 54.4 | 602 | 41 | 0.66/0.10/0.24 |

size of 256 was used for the encoder, GNN and decoder layers. Additionally, the best hyperparameters corresponding to each experiment to reproduce the results in the main paper are reported in the tables below.

$lr$ refers to the learning rate, $n_{enc}$ refers to the number of layers in the encoder MLP, $n_{dec}$ refers to the number of layers in the decoder MLP, $\lambda$ refers to the noise multiplier, and $K$ refers to the maximum degree.

Table 9: **Hyperparameters for models in Table 1.**

| Model | Dataset | $lr$ | Batch Size | Activation | $n_{enc}$ | $n_{dec}$ | $\lambda$ | $K$ |
|---|---|---|---|---|---|---|---|---|
| GCN | ogbn-arxiv | 0.002 | 1,000 | ReLU | 2 | 2 | - | 30 |
|  | ogbn-products | 0.001 | 1,000 | ReLU | 2 | 1 | - | 30 |
|  | ogbn-mag | 0.001 | 1,000 | ReLU | 2 | 2 | - | 10 |
|  | reddit | 0.001 | 1,000 | ReLU | 2 | 2 | - | 10 |
| DP-GCN (Adam) | ogbn-arxiv | 0.003 | 10,000 | Tanh | 1 | 2 | 2 | 7 |
|  | ogbn-products | 0.005 | 10,000 | Tanh | 1 | 2 | 1 | 10 |
|  | ogbn-mag | 0.001 | 10,000 | Tanh | 2 | 2 | 2 | 10 |
|  | reddit | 0.002 | 10,000 | Tanh | 2 | 2 | 1 | 10 |
| DP-GCN (SGD) | ogbn-arxiv | 1.0 | 10,000 | Tanh | 2 | 1 | 2 | 7 |
|  | ogbn-products | 2.0 | 40,000 | Tanh | 1 | 1 | 2 | 5 |
|  | ogbn-mag | 1.0 | 10,000 | ReLU | 1 | 2 | 1 | 10 |
|  | reddit | 0.1 | 10,000 | Tanh | 1 | 2 | 1 | 10 |
| MLP | ogbn-arxiv | 0.001 | 1,000 | ReLU | 2 | 1 | - | - |
|  | ogbn-products | 0.001 | 1,000 | ReLU | 1 | 2 | - | - |
|  | ogbn-mag | 0.01 | 1,024 | ReLU | 2 | 1 | - | - |
|  | reddit | 0.001 | 1,000 | ReLU | 1 | 1 | - | - |
| DP-MLP | ogbn-arxiv | 0.003 | 10,000 | Tanh | 1 | 2 | 1 | 0 |
|  | ogbn-products | 0.002 | 10,000 | ReLU | 1 | 2 | 1 | 10 |
|  | ogbn-mag | 0.001 | 10,000 | ReLU | 2 | 2 | 1 | 10 |
|  | reddit | 0.001 | 10,000 | Tanh | 1 | 2 | 1 | 10 |

---

[6]Indices indicating the split for each node available at this Google Drive link. '0' indicates the train split, '1' indicates the validation split, and '2' indicates the test split.

Table 10: **Hyperparameters for models in Table 2.**

| Model | Dataset | $lr$ | Batch Size | Activation | $n_{\text{enc}}$ | $n_{\text{dec}}$ | $\lambda$ | $K$ |
|---|---|---|---|---|---|---|---|---|
| | ogbn-arxiv-disjoint | 0.001 | 1,000 | ReLU | 1 | 1 | - | 30 |
| GCN (2-layer) | ogbn-arxiv-clustered | 0.001 | 1,000 | ReLU | 1 | 2 | - | 10 |
| | reddit-disjoint | 0.001 | 1,000 | ReLU | 2 | 2 | - | 10 |
| | ogbn-arxiv-disjoint | 0.001 | 1,000 | ReLU | 1 | 2 | - | 7 |
| GCN (1-layer) | ogbn-arxiv-clustered | 0.001 | 1,000 | ReLU | 1 | 1 | - | 30 |
| | reddit-disjoint | 0.001 | 1,000 | ReLU | 2 | 2 | - | 10 |
| | ogbn-arxiv-disjoint | 0.003 | 20,000 | Tanh | 1 | 1 | 2 | 3 |
| DP-GCN (2-layer) | ogbn-arxiv-clustered | 0.002 | 20,000 | Tanh | 2 | 1 | 4 | 3 |
| | reddit-disjoint | 0.002 | 20,000 | Tanh | 1 | 1 | 2 | 3 |
| | ogbn-arxiv-disjoint | 0.002 | 20,000 | Tanh | 1 | 2 | 4 | 7 |
| DP-GCN (1-layer) | ogbn-arxiv-clustered | 0.002 | 20,000 | Tanh | 2 | 2 | 4 | 10 |
| | reddit-disjoint | 0.001 | 10,000 | Tanh | 2 | 2 | 1 | 10 |
| | ogbn-arxiv-disjoint | 0.001 | 1,000 | ReLU | 2 | 1 | - | - |
| MLP | ogbn-arxiv-clustered | 0.002 | 1,000 | ReLU | 1 | 2 | - | - |
| | reddit-disjoint | 0.001 | 1,000 | ReLU | 1 | 1 | - | - |
| | ogbn-arxiv-disjoint | 0.003 | 10,000 | Tanh | 1 | 2 | 1 | 10 |
| DP-MLP | ogbn-arxiv-clustered | 0.003 | 10,000 | Tanh | 2 | 2 | 1 | 10 |
| | reddit-disjoint | 0.001 | 10,000 | Tanh | 1 | 2 | 1 | 10 |

Table 11: **Hyperparameters for models in Table 3.**

| Architecture | Method | $lr$ | Batch Size | Activation | $n_{\text{enc}}$ | $n_{\text{dec}}$ | $\lambda$ | $K$ |
|---|---|---|---|---|---|---|---|---|
| GCN | Non-Private GNN | 0.002 | 1,000 | ReLU | 2 | 2 | - | 30 |
| | DP-GNN | 0.003 | 10,000 | Tanh | 1 | 2 | 2 | 7 |
| GIN | Non-Private GNN | 0.003 | 1,000 | ReLU | 2 | 1 | - | 30 |
| | DP-GNN | 0.003 | 10,000 | Tanh | 1 | 1 | 1 | 7 |
| GAT | Non-Private GNN | 0.001 | 1,000 | Tanh | 2 | 2 | - | 30 |
| | DP-GNN | 0.004 | 20,000 | Tanh | 1 | 2 | 2 | 7 |
| MLP | Non-Private MLP | 0.001 | 1,000 | ReLU | 2 | 1 | - | - |
| | DP-MLP | 0.003 | 10,000 | Tanh | 1 | 2 | 1 | - |

Table 12: **Hyperparameters for models in Table 4.**

| Model | Dataset | $lr$ | Batch Size | Activation | $n_{\text{enc}}$ | $n_{\text{dec}}$ | $\lambda$ | $K$ |
|---|---|---|---|---|---|---|---|---|
| | ogbn-arxiv-disjoint | 0.001 | 1,000 | ReLU | 1 | 2 | - | 3 |
| GIN (2-layer) | ogbn-arxiv-clustered | 0.001 | 1,000 | ReLU | 2 | 2 | - | 30 |
| | reddit-disjoint | 0.001 | 1,000 | ReLU | 2 | 2 | - | 10 |
| | ogbn-arxiv-disjoint | 0.002 | 1,000 | ReLU | 2 | 1 | - | 30 |
| GIN (1-layer) | ogbn-arxiv-clustered | 0.001 | 1,000 | ReLU | 2 | 1 | - | 30 |
| | reddit-disjoint | 0.001 | 1,000 | ReLU | 2 | 1 | - | 10 |
| | ogbn-arxiv-disjoint | 0.002 | 20,000 | Tanh | 2 | 2 | 2 | 3 |
| DP-GIN (2-layer) | ogbn-arxiv-clustered | 0.002 | 10,000 | Tanh | 1 | 2 | 1 | 3 |
| | reddit-disjoint | 0.001 | 10,000 | Tanh | 1 | 1 | 1 | 3 |
| | ogbn-arxiv-disjoint | 0.004 | 20,000 | Tanh | 1 | 2 | 2 | 7 |
| DP-GIN (1-layer) | ogbn-arxiv-clustered | 0.004 | 10,000 | Tanh | 1 | 1 | 1 | 10 |
| | reddit-disjoint | 0.001 | 10,000 | Tanh | 2 | 2 | 1 | 10 |
| | ogbn-arxiv-disjoint | 0.001 | 1,000 | ReLU | 2 | 1 | - | 10 |
| MLP | ogbn-arxiv-clustered | 0.002 | 1,000 | ReLU | 1 | 2 | - | 10 |
| | reddit-disjoint | 0.001 | 1,000 | ReLU | 1 | 1 | - | 10 |
| | ogbn-arxiv-disjoint | 0.003 | 10,000 | Tanh | 1 | 2 | 1 | 10 |
| DP-MLP | ogbn-arxiv-clustered | 0.003 | 10,000 | Tanh | 2 | 2 | 1 | 10 |
| | reddit-disjoint | 0.001 | 10,000 | Tanh | 1 | 2 | 1 | 10 |

