# OpenReview forum: "Node-Level Differentially Private Graph Neural Networks"
_TMLR — Withdrawn by Authors_

### Review · Reviewer_VQxK · 2023-01-08

**Summary Of Contributions:**

This paper proposes an algorithm that guarantees the node-level (for both node feature and adjacency list) privacy. This algorithm only guarantees privacy during training, not during inference.

**Audience:**

Yes

**Claims And Evidence:**

Yes

**Requested Changes:**

	I understand that the problem studied is hard. The author should discuss how to protect privacy when doing inference.



**Strengths And Weaknesses:**

Strengths: Novel solution and solid theoretical analysis
	Weaknesses:
not scalable to GNN with deeper layers;
not applicable for privacy protection when doing inference (which still incurs queries on the adjacency list), and this may be a problem because, in the Transductive setting, sensitive nodes will be used (for prediction on itself or other nodes).
 Privacy seems relatively weak (epsilon > 10).

---

### Review · Reviewer_Kvcc · 2023-01-16

**Summary Of Contributions:**

This paper proposes a new method for learning graph neural networks (GNNs) under differential privacy (DP). Authors consider both the nodes and edges of a graph as sensitive, and the proposed method is aimed to satisfy what authors call node-level DP. In this trust model, the adjacency is considered to extend to both the nodes and edges of the graph. Several crucial components of standard DP-SGD do not extend to this privacy definition (e.g. sensitivity and subsampling amplification), and authors suggest novel techniques to replace these parts. The main result is the DP-GNN method (Algorithm 4). Empirical results demonstrate that the proposed method is able to learn better predictive models than a DP-SGD applied on neural networks that do not use the graph information.

**Audience:**

Yes

**Broader Impact Concerns:**

I don't think the paper needs a broader impact statement.

**Claims And Evidence:**

Yes

**Requested Changes:**

I will list below some clarifications that would be necessary/beneficial for the paper.

- Clarification: Regarding the "Gradient clipping" paragraph. The sensitivity term seems to be now $2C \cdot N(K, r)$. However, you state that you use layer-wise clipping. So is $C$ now the maximum gradient norm of the full, layer-wise clipped, gradient? Or is the privacy budget somehow split across different layers? Also, is there some privacy-preserving protocol used to choose the 75th norm percentile for the gradient clipping?
- Clarification: About the use of $\leq \epsilon$ in the results. For Tables 1-6, you have used $\leq \epsilon$ to denote the privacy budget. Can you clarify a bit what this means? Are all the results you show for the DP methods still trained with the same budget?
- Clarification: The proof for Lemma 1. You state that if $S_{u'} \in S^{r+1}(v)$, then $S_{u'} \in S^{r}(v)$. I thought that $S^{r+1}(v)$ would be larger set than $S^{r}(v)$, and therefore the claim looks odd. However, it is completely possible I missed something important that would make the claim hold, but it might make sense to unpack these proofs a bit.
- Clarification: Add explanation for the $\pm$ used in Tables.
- Clarification: Discuss how your node-level DP differs from the existing definitions of same name, or alternatively add reference to work from which your Definitions 1 and 2 come from.
- Suggestion: For audience less familiar with DP, it might make sense to add a brief description of the Clip function (for example in the appendix).

**Strengths And Weaknesses:**

Strengths:

- Novel algorithm and trust model for private learning of GNNs
- Compared to earlier works considering node-level DP, the proposed method operates in centralized setting compared to local DP
- Empirical results (on rather large $\epsilon$) demonstrate that the DP overhead arising from more complex neighbouring relationship is smaller than the gain from using the graph information for the prediction.

Weaknesses:

- The paper is rather dense, and somewhat difficult to read for someone without background on GNNs (like me)
- As far as I understand, the node-level DP (at least as a name) is not a novel concept. Hence it would be important for authors to clearly state if the Definitions 1 and 2 are novel privacy models, or if they are based on existing work.
- The empirical validation could be better described. For example, are the results shown a single DP run results or are there more repetitions? What are the errors reported in Tables? What does $\epsilon \leq 12$ mean, are the different methods comparable in terms of privacy?

---

### Review · Reviewer_zTeg · 2023-01-30

**Summary Of Contributions:**

This paper studies the problem of differentially private graph neural networks. More specifically, the authors consider the node-level privacy guarantee and develop novel training algorithms to achieve strong privacy and utility guarantees. Experimental results also validate the effectiveness of the proposed methods.

**Audience:**

Yes

**Claims And Evidence:**

Yes

**Requested Changes:**

The following issues need to be addressed in the current paper:
1. There is no discussion about the proposed algorithms 1,2,3. The authors need to give a simple description of these algorithms (key idea, goal, key innovation and etc.).
2. What is the computational complexity of the proposed algorithm? Whether it is efficient or not? For example, what is the computational cost of Algorithm 3 and the clipping procedure? How does it compare to the original method?
3. In your Theorem 1, what is the constraint on $\alpha$? In the subsample amplification results, there is often a constraint on the parameter $/alpha$.
4. What will happen to Theorem 1 if you use all the data?
5. In your experiments, whether the baseline GCN method is the common GCN method or the one uses the subsample data?
6. In your ablation study, you only consider the high $\epsilon$ case. What about the small $\epsilon$?
7. I'm curious about the training time and memory costs for the private method and the non-private one.
8. What is the iteration number $T$ for different methods?
9. When you choose the clipping parameter, it seems that you are using the gradient norm information, which is non-private.


**Strengths And Weaknesses:**

Strengths:
1. The problem considered here is very interesting and relevant
2. The proposed algorithms seem to be novel
3. Experimental results validate the effectiveness of the proposed algorithms

Weakness:
1. The presentation needs to be improved (see requested changes part)
2. It seems that the algorithm works well only when the privacy budget $\epsilon$ is very high
3. More evaluations are needed to justify the proposed methods (see requested changes part)

---

### Note · Authors · 2023-02-14

**Comment:**

We realized an honest error in our implementation regarding the computation of clipping thresholds, which affects the empirical results discussed in our paper. We wish to withdraw this paper and resubmit once these issues are fixed.
We apologize again for all of the inconvenience caused to the Action Editor and reviewers.

**Withdrawal Confirmation:**

I have read and agree with the venue's withdrawal policy on behalf of myself and my co-authors.